# Modelling and Validation of a Guided Acoustic Wave Temperature Monitoring System

**DOI:** 10.3390/s21217390

**Published:** 2021-11-06

**Authors:** Lawrence Yule, Bahareh Zaghari, Nicholas Harris, Martyn Hill

**Affiliations:** 1Electronics and Computer Science, University of Southampton, Southampton SO17 1BJ, UK; nrh@ecs.soton.ac.uk; 2School of Aerospace, Transport and Manufacturing, Cranfield University, Bedford MK43 0AL, UK; bahareh.zaghari@cranfield.ac.uk; 3School of Engineering, University of Southampton, Southampton SO17 1BJ, UK; m.hill@soton.ac.uk

**Keywords:** condition monitoring, guided waves, COMSOL, wedge transducer, nozzle guide vane

## Abstract

The computer modelling of condition monitoring sensors can aide in their development, improve their performance, and allow for the analysis of sensor impact on component operation. This article details the development of a COMSOL model for a guided wave-based temperature monitoring system, with a view to using the technology in the future for the temperature monitoring of nozzle guide vanes, found in the hot section of aeroengines. The model is based on an experimental test system that acts as a method of validation for the model. Piezoelectric wedge transducers were used to excite the $S_{0}$ Lamb wave mode in an aluminium plate, which was temperature controlled using a hot plate. Time of flight measurements were carried out in MATLAB and used to calculate group velocity. The results were compared to theoretical wave velocities extracted from dispersion curves. The assembly and validation of such a model can aide in the future development of guided wave based sensor systems, and the methods provided can act as a guide for building similar COMSOL models. The results show that the model is in good agreement with the experimental equivalent, which is also in line with theoretical predictions.

## 1. Introduction

Structural health monitoring (SHM) is defined as damage identification within structures, systems, and components, using permanently installed sensors that monitor changes over time [1]. This builds upon techniques utilised in nondestructive evaluation (NDE). A wide range of structural and mechanical systems benefit from SHM in industries such as aerospace, civil engineering, and automotive. In general monitoring is carried out using unobtrusive sensors that do not interfere with the use or operation of the chosen structure. Installing such systems can reduce the need for regular maintenance schedules, which reduces costs and can improve safety and reliability by identifying problems before they become dangerous [2]. SHM can be classified into two groups, passive or active, where passive monitoring does not utilise an external signal (acoustic emission [3] and strain sensors [4]), and active monitoring excites the structure of interest and analyses the transmitted signal. Active systems often use piezoelectric sensors to excite and detect guided waves [5,6]. In many implementations of SHM, the systems rely on comparisons with safe/healthy baselines to determine whether damage has occurred, and in some cases, rely on prior knowledge of material properties, operational conditions, or data from other monitoring systems. One of the many challenges in successfully utilising SHM systems is operating sensors under harsh operational or environmental conditions, which can have an effect on the longevity of the sensors, as well as increase the complexity of analysing the collected data. Applying machine-learning techniques to these complex data sets is an emerging technology that can be used to identify changes that would otherwise be difficult to recognise using traditional analysis methods [7,8]. A review of SHM techniques is provided by Tibaduiza Burgos et al. [9].

The advent of small robust sensors has allowed considerable amounts of monitoring to take place, but there is still scope to extend monitoring to harsher environments. The monitoring of aerospace components at high temperatures is of particular interest as operating components closer to their thermal limits can increase efficiency, which can reduce operational costs [10]. Exposure to high temperatures increases the risk of failure, which makes health monitoring of these components vitally important. During the design stage, finite element models are often used to predict thermal stresses but are difficult to verify experimentally [11]. Additional high temperature sensors can feed into active control systems to ensure optimal operation of complex systems. The development of new high temperature sensors can inform design decisions and improve the accuracy of models, which will lead to increased efficiency in the finalised component. The modelling of sensor systems as well as the components themselves can enable the impact of the sensors on component operation to be analysed.

### 1.1. Ultrasonic Structural Health Monitoring

Ultrasound is of particular interest for structural health monitoring as it can be utilised for a large number of applications, such as pipe [12] and rail inspection methods [13] or defect detection for aircraft [14]. Traditional ultrasonic nondestructive evaluation (NDE) techniques utilise A-scans, a measurement of signal amplitude against time, to detect cracks, defects, etc. This can be extended to B [15], C [16,17], or phased-array [18] scans to build an image of damage in an area by moving the transducers around and carrying out multiple measurements. Surface acoustic waves (SAWs) are also used for structural health monitoring applications, commonly based on the use of interdigital transducers (IDTs) operated as either delay lines or resonators, which can operate in harsh environments and can be interrogated wirelessly [19,20]. These types of sensor generally propagate SAWs within their own structure, rather than through the material on which they are deposited. In order to propagate a wave through an existing structure and over long distances, Rayleigh and Lamb waves can be used. Rayleigh waves are suited to a number of structural health monitoring applications as they are highly sensitive to any discontinuities, defects, or surface coatings [21]; however, this limits their use for certain applications (such as temperature monitoring).

Lamb waves are “guided” by the upper and lower boundaries of a material allowing for continuous wave propagation [22]. They can travel large distances with limited attenuation using constructive interference with surfaces/boundaries. Unlike bulk acoustic waves, Lamb waves are dispersive and multimodal, which makes their analysis complex, especially when there are other factors such as changing temperatures involved. The lowest order modes, the fundamental antisymmetric mode $A_{0}$, and the fundamental symmetric mode $S_{0}$, are the most commonly used modes as they are relatively nondispersive and comparatively easy to generate in comparison to the higher order modes ($A_{1}$, $S_{1}$, etc.). Lower order Lamb waves are used extensively for NDE and SHM applications, and an overview of their uses for damage identification is provided by Su [23]. Lamb waves have both phase and group velocities, the phase velocity relating to the local speed with which phase of the wave changes, and a group velocity that describes the overall speed of energy transport through the propagating wave. Phase velocity is generally higher than the group velocity. Time of flight ($t_{F}$) measurements of Lamb waves give the group velocity, while special phase comparison techniques are needed to measure the phase velocity [24].

### 1.2. Guided Wave Temperature Monitoring

Despite the numerous uses of guided waves, they have had very limited use for temperature monitoring applications. However, the fundamental antisymmetric Lamb wave mode, $A_{0}$, has been used for temperature monitoring of silicon wafers during rapid thermal processing [25,26]. Quartz pins are used as waveguides, connecting to the wafer through Hertzian contact points. Time of flight ($t_{F}$) was measured at a rate of 20 Hz from 100 ${}^{\circ }C$ to 1000 ${}^{\circ }C$ with an accuracy of ±5 ${}^{\circ }C$ with this method. A laser excitation system has also been used to measure the temperature of silicon wafers during rapid thermal processing [27]. These examples show the potential of a guided wave-based temperature monitoring system. Sensors can be placed away from harsh environments, and the operation of the monitoring system will not impact the operation of the device/component itself. Working at high frequencies can allow for fast response times and resolutions/accuracy comparable with traditional temperature sensors. The basis of developing a temperature monitoring system using ultrasonic guided waves relies on temperature having an effect on wave propagation. Any change in material properties within the propagation medium will have an effect on wave propagation. A change in temperature causes a change in the Young’s modulus, Poisson’s ratio, and density. Young’s modulus has the largest effect on wave propagation, reducing with increasing temperature, which causes a reduction in wave velocity. Density also decreases with increasing temperature, manifesting as thermal expansion, which has a relatively small effect on wave velocity. Poisson’s ratio changes can have a large impact on wave propagation but generally only occur over large temperature ranges [28]. As temperature change causes a change in wave velocity, time of flight measurement can be used to monitor a change in temperature. Group velocity can be calculated if the propagation distance is precisely known, which can be compared to theoretical values to measure temperature directly. This is the basis for the monitoring system described in this paper.

### 1.3. Potential Applications of an Ultrasonic Temperature Monitoring System

An example of a potential application for this technology is nozzle guide vanes (NGVs), which are static components located in the turbine section of jet engines that are operated at extremely high temperatures (up to 1800 ${}^{\circ }C$ with cooled blades [29]). The monitoring of these components is challenging because of their location and the extreme temperatures and gas pressures that they are exposed to. There are a number of well established offline monitoring methods (thermal paints and thermal history sensors) that record the peak temperature of exposure during an operational cycle of a turbine but considerably less well established methods for online monitoring during normal operation (thin film thermocouples, thermographic phosphors, and pyrometers). The offline methods are used in the design stage to verify analytical models, locate areas of the component with high temperature gradients, and ensure that the component does not exceed temperature limits. Online methods can provide considerably more data during start-up and shut-down of the engine, as well as recording any overshoot events. Online methods can be used for health monitoring and data can be fed into active control systems. A comprehensive review of both offline and online monitoring methods and an introduction to ultrasonic based temperature monitoring is provided by Yule et al. [30].

The technique could also be applied to any structure where ultrasonic waves will propagate. Batteries that are made up of a number of cells could be monitored using this method to detect a failing cell before it becomes dangerous. Printed circuit boards (PCBs) can contain many components that reach high temperatures; an ultrasonic temperature monitoring method could reduce the number of sensors required to sample the board and reduce the impact of the sensors on operation. The system could be used in combination with baseline comparisons and machine learning techniques to analyse complex wave packets and detect changes in the response due to temperature.

The following sections describe the development of a COMSOL model based on an experimental test system used to monitor temperature using the propagation of ultrasonic Lamb waves. A COMSOL model has been developed to simulate guided wave propagation in an aluminium plate, where the environment can easily be adjusted to evaluate the impact on wave propagation and sensor operation. A guide to building and running the model is provided, along with validation of the model against theoretical Lamb wave temperature sensitivity extracted from dispersion curves, and experimental results from the test system that the COMSOL model replicates.

## 2. Lamb Wave Mode Targeting

The dispersive multimodal nature of Lamb waves means that careful selection of excitation frequency is required to target wave modes that can be more easily analysed. The choice of frequency/mode also determines the temperature sensitivity. The generation of dispersion curves based on material properties aides in this process, and they can be used to validate the results of simulated or experimental measurements. The $S_{0}$ Lamb wave mode was targeted at a frequency-thickness product of 1 MHz-mm (in a 1 mm plate), as shown in Figure 1. Theoretical dispersion curves calculated from the material properties of aluminium (see Table 1) were produced using The Dispersion Calculator [31], a free software for calculating dispersion curves of guided waves. Group velocity curves were generated at 10 ${}^{\circ }C$ intervals from 10 to 110 ${}^{\circ }C$ by varying Young’s modulus (see Equation (Equation 4)), and the velocity at 1 MHz-mm was extracted from each curve. It should be noted that the material properties provided by Hopkins refer to aluminium in general, and not specifically the aluminium 1050 H14 that was used in the experimental section of this study. The velocities extracted from dispersion curves are plotted against experimental and simulated results in Section 6.

Using wedge transducers allows the targeting of single Lamb wave modes with careful selection of excitation angle. The angle is determined based on Snell’s law:(1)$$\mathrm{Angle}\;\theta ={\mathrm{Sin}}^{-1}\left(\frac{\mathrm{Longitudinal}\;\mathrm{wedge}\;\mathrm{velocity}}{\mathrm{Lamb}\;\mathrm{wave}\;\mathrm{phase}\;\mathrm{velocity}}\right)$$

The wedge angle required to excite the $S_{0}$ mode is:(2)$$31{}^{\circ }={\mathrm{Sin}}^{-1}\left(\frac{2720}{5258}\right)$$

The $A_{0}$ mode cannot be excited using this method as the phase velocity at this frequency (2312 m s${}^{-1}$) is slower than the longitudinal velocity of the wedge. If the $A_{0}$ mode is present in the signal it will not affect measurement of the $S_{0}$ mode as its group velocity is significantly different than that of the $S_{0}$ mode, which will cause a distinct second wave packet.

It should be made clear, however, that the use of wedge transducers is unlikely to be the finalised transducer configuration for a permanently installed temperature monitoring system, as they rely on liquid couplants and acrylic wedges that would melt at relatively low temperatures (∼150 ${}^{\circ }C$). The use of wedges at this stage were useful for simplifying the signal processing techniques required, and targeting specific wave modes of interest. For permanent installation there are a number of options available. Piezoelectric Wafer Active Sensors (PWAS) are being used extensively for SHM applications and have been shown to withstand exposure to extreme environments [32]. They are nonresonant wide-band devices; however, they can be used for generation of single Lamb wave modes with careful geometry selection [33]. PWAS are small, inexpensive, and minimally invasive [34], making them potentially suitable for installation on NGVs if a suitable bonding method and piezoelectric material can be found. Another solution to this problem is to couple them to the structure using wave guides and Hertzian contact points, which would allow the transducers to be placed further away from the harsh environment. This method of coupling has been used to measure the mechanical properties of carbon fibre reinforced plastics (CFRP) using measured phase velocities of the $A_{0}$ and $S_{0}$ Lamb wave modes [35].

## 3. COMSOL Simulation Methods

The multiphysics simulation package COMSOL has been used to simulate a potential temperature monitoring system, as described experimentally in Section 4. The experimental measurement system was used as validation of the COMSOL model. The literature covering the use of COMSOL for modelling Lamb wave excitation using wedge transducers is limited; however, it has been shown that Lamb waves can be successfully generated using this method [36].

The model consisted of two variable angle wedges (PMMA), which were based on the geometry of Olympus variable angle wedges, placed on top of an aluminium plate. The thickness of the plate was set to 1 mm to target the $S_{0}$ mode at 1 MHz–mm. The transmitting wedge had a simplified piezoelectric transducer (PZT-5H from COMSOL’s material library) attached to its rotating block, to which the excitation signal was applied. The rotating block was set to an angle of 31${}^{\circ }$. The geometry can be seen in Figure 2. The received signal was measured at the receiver wedge’s rotating block boundary. More realistic transducer configurations were not considered in this study, as the focus was on the effect of temperature on the propagating wave. A boundary area was set underneath the plate to act as the heat source, again mimicking the experimental setup. This was simplified to allow the temperature to be directly set, rather than simulating a hot plate.

The change in Young’s Modulus with temperature was included in the material properties for both the wedges (Equation (Equation 3)) [37] and the aluminium (Equation (Equation 4)) [38] using piecewise functions.
(3)$$E\left(T\right)=\;-15250\times T^{2}+1125000\times T+4932500000\;$$
(4)$$E\left(T\right)=\;-4\times 10^{7}\times T+8\times 10^{10}\;$$
where *T* is the temperature in Kelvin. The change in Poisson’s ratio is considered negligible over this temperature range [28] and was not included in the simulation. Thermal expansion is also considered to have a negligible effect on the propagation distance (and change in density) and was excluded (calculated to have an average reduction in wave velocity of the $S_{0}$ mode in aluminium of −1.20 m s${}^{-1}$ over the temperature range 20–100 ${}^{\circ }C$). The modules Solid Mechanics, Electrostatics, and Heat Transfer in Solids were used in this simulation, along with a multiphysics node to couple Solid Mechanics with Electrostatics for the piezoelectric effect. Both the wedges and the plate were set to isotropic linear elastic materials, with low reflecting boundaries applied to the wedges.

The simple piezoelectric transducer for the transmitting wedge was set up as follows: A zero charge node was used for the edges of the material, initial values were set to 0 V, a “Charge Conservation, Piezoelectric” node was set for the material, a ground boundary was selected for the wedge side of the material, and a terminal node was set for the opposite boundary. Within the terminal node the type was set to Voltage and the input was set to V0(t). The excitation signal was a 1 MHz 5–cycle Hamming windowed sine pulse generated in MATLAB and imported into COMSOL using linear interpolation (Definitions > Interpolation).

For the Heat Transfer in Solids module all the domains were set to solid, and initial values were set to 20 ${}^{\circ }C$. The boundaries exposed to the air were selected in a Heat Flux node, where convective heat flux was selected. A user defined heat transfer coefficient of 15 W/(m${}^{2}\;\cdot \;$K) was used for the plate and 5 W/(m${}^{2}\;\cdot \;$K) for the wedges. These values were adjusted to produce the temperature gradients measured experimentally in both the plate and the wedges. The external temperature was set to 20 ${}^{\circ }C$. The temperature of the boundary underneath the plate was adjusted as required (20 ${}^{\circ }C$ to 100 ${}^{\circ }C$ in 20 ${}^{\circ }C$ increments for this study). An example of the temperature gradients produced from the stationary study step is shown in Figure 3, where the temperature boundary underneath the plate was set to 100 ${}^{\circ }C$.

The mesh size for each material was determined by excitation frequency. The excitation wavelength for each of the materials was calculated by dividing their longitudinal wave speed by $f_{0}$. A free triangular mesh was created for each of the materials, and the maximum element size for each of them was set to LocalWavelength/N. If higher frequency content is expected, the wavelength for each material should be based on the highest frequency expected rather than $f_{0}$. In order to accurately resolve a wave, at least 10–12 elements per local wavelength are required [39]. This assumes linear discretization for all modules. Using 12 elements results in an average skewness rating (measure of element quality, 0–1) of 0.9345 over 154,728 elements [40]. This is equivalent to a sample rate of 1.2 × 10${}^{8}$.

This study had two steps, firstly, a stationary study to simulate the effect of temperature on the system until an equilibrium was reached, and secondly, a time dependent study to simulate wave propagation that had its initial conditions set by the stationary study. The settings for the initial study were adjusted to solve for heat transfer but not for electrostatics/the piezoelectric effect. Changing temperature causes a change in Young’s modulus, which subsequently affects wave velocity.

The time dependent study included electrostatics/the piezoelectric effect to allow for wave generation but did not include heat transfer. This reduced the computation time as it was not necessary to model changing temperature as the time dependent model solved, only to use the fixed values of Young’s modulus that were passed on from the stationary study. The time dependent study had its “Output times” set to: range(0, dt, sim_length) where “dt” is a global definition parameter equal to CFL/(N $\times \;f_{0}$). The CFL (Courant Friedrichs Lewy) number is suggested by COMSOL [41] to be less than 0.2, optimally 0.1 (when the default second order quadratic mesh elements are used). This value represents the relationship between wave speed, *c*, maximum mesh size, *h*, and time step length, Δt: CFL=cΔt/h. This can be rewritten in terms of frequency, as the maximum mesh size *h* has already been manually defined by *N*, the number of elements per local wavelength for each material: CFL=fNΔt. This can then be rearranged to give the time step: Δt=CFL/Nf.

Under “Values of Dependent Variables”, the settings were changed to user controlled, the method was changed to Solution, and the study was set to the stationary study. The time step was manually set under Solver Configurations > Solution 1 > Time dependent solver > Time stepping. Here, the “Steps taken by solver” parameter was changed to “Manual” and the “Time Step” was set to: $CFL/(N\times f_{0})$.

To reduce file size, only the data at the wedge boundaries was stored by the solver. This was achieved by adding an “Explicit Selection” node in the Geometry section, and selecting both the transmit and receive wedge boundaries. Within the time dependent study settings, we selected “For selection” under “Store fields in output” and selected the boundary group [42].

A parametric sweep node was used to cycle through the temperature boundary values (20 ${}^{\circ }C$ to 100 ${}^{\circ }C$ in 20 ${}^{\circ }C$ increments) and save the output of the time dependent model for each value. This was repeated for the model in the wedge-to-wedge configuration (mimicking the experimental setup shown in Section 4). The simulations were run on the University of Southampton’s IRIDIS 5 supercomputing platform [43].

Exaggerated deformation of pressure in the plate as seen in Figure 4 made the presence of the $A_{0}$ and $S_{0}$ modes clearly visible. The modes were separated in the time domain after a short distance (∼50 mm) due to the difference in group velocity. To visualise wave propagation and calculate time of flight the pressure at both transmitter and receiver wedge boundaries were exported. An example of wave propagation at room temperature can be seen in Figure 5, where the thick blue and orange lines indicate the envelope of the signals, while the blue dotted lines indicate the peak of the envelopes. These points were used to calculate the time-of-flight.

The next section covers the experimental temperature measurement system upon which the COMSOL model was based and that was used to validate the model. The method for calculating time of flight for both systems is covered in Section 5.

## 4. Experimental Study

Two 1 MHz piezoelectric transducers (12 mm diameter) attached to acrylic (PMMA) wedges (Olympus variable angle wedge) in a pitch-catch configuration were coupled to a 1 mm thick aluminium plate (1050 H14) with a liquid couplant (Figure 6). The wedges had a maximum operating temperature of ∼150 ${}^{\circ }C$, while the couplant had a maximum operating temperature of ∼100 ${}^{\circ }C$. A signal generator (GW Instek MFG-2203M) was used to generate a 5-cycle Hamming windowed tone burst at 1 MHz. Signals were digitised using a Picoscope 3406DMSO USB Oscilloscope. The hardware used in this study is given in Table 2. Based on a sampling rate of 5 × 10${}^{8}$ Hz, the theoretical maximum temporal resolution was 2 ns. Signal processing was carried out in MATLAB. A zero-phase bandpass filter was applied to the signals to remove unwanted noise. The time of flight ($t_{F}$) was measured between transducers, and wave velocity was calculated from the distance between transducers. The temperature of the aluminium plate was controlled using a hot plate.

The hot plate was used to raise the temperature of the aluminium plate to the desired temperature. The temperature of the aluminium plate was monitored using a thermocouple placed in the centre of the plate at the hottest point. The total $t_{F}$ was measured until it stabilised, using the test setup shown in Figure 7. The temperature of the entire system was allowed to stabilise before a measurement was taken. The wedges were aligned using a 3D printed spacer to ensure that the distance between them was 100 mm. Multiple measurements were taken after adjusting both wedge positions. The wedges were removed from the surface and placed together to measure the wedge-to-wedge $t_{F}$, as shown in Figure 8. Multiple measurements were taken after adjusting the wedge-to-wedge position. The $t_{F}$ measurement process was repeated after allowing the total $t_{F}$ to restabilise. The velocity was calculated using Equation (Equation 9). The mean average was calculated from the results of the repeated total $t_{F}$ measurements, and the velocity was calculated for every wedge-to-wedge result. The average velocity was calculated along with the standard deviation. An example of wave propagation at room temperature can be seen in Figure 9, where the thick blue and orange lines indicate the envelope of the signals, while the blue dotted lines indicate the peak of the envelopes. These points were used to calculate the time-of-flight.

During the measurement of total $t_{F}$, the temperature of the plate was measured using a single thermocouple placed in the centre of the transmission path. After the total $t_{F}$ measurements were completed, the wedges were removed from the plate, and additional thermocouples (4) were placed along the transmission path (centrally, 30 mm, 60 mm, and 90 mm along) to measure the temperature gradient up to the back side of a wedge. Measurements were repeated after moving the thermocouples to the other half of the transmission path. The temperature of the hot plate was set so that it matched the temperature measured from the central thermocouple during measurement of total $t_{F}$. The mean average temperature was calculated for the total transmission path at each hot plate temperature setting.

## 5. Velocity Calculation

Calculation of wave velocity depends on measurement of time of flight ($t_{F}$), which can be described by Equation [44]:(5)$$t_{F}=\frac{d}{c}$$
where *d* is the distance travelled at wave speed *c*, both of which are functions of temperature, *T*. The sensitivity of the time of flight to temperature can then be expressed as:(6)$$\delta t_{F}=\frac{d}{c}\left(\alpha -\frac{k}{c}\right)\delta T$$
where α is the coefficient of thermal expansion of the medium and *k* is the rate of change of wave velocity with temperature:(7)$$k=\frac{\delta c}{\delta T}$$

Time of flight was calculated using the same method for both the COMSOL model and the experimental system. The analytic envelope was computed for both the excitation signal and the received signal, the difference in time between the peaks of these envelopes was taken as the time of flight. This method of time of flight measurement can also be applied to more dispersive signals, which cannot be achieved using cross-correlation methods. Various signal processing techniques for time of flight measurement are discussed in detail by Guers [45]. Group velocity was then calculated using Equations (Equation 8) and (Equation 9). The propagation time through the wedges (measured using the configuration shown in Figure 8) was subtracted from the total $t_{F}$ to ensure that only the propagation time through the plate was measured. The distance between wedges for both studies was 100 mm.
(8)$$v=\frac{d}{t_{F}}$$
(9)$$v=\left(\frac{d\;\mathrm{between}\;\mathrm{wedges}+d\;\mathrm{wedge}\;\mathrm{foot}\;\mathrm{offset}}{\mathrm{Total}\;t_{F}-\mathrm{Wedge}-\mathrm{to}-\mathrm{wedge}\;t_{F}}\right)$$
where the *d* wedge foot offset is an unknown distance from the front edge of the wedge to where the wave enters the plate from the wedge. For the experimental system, this distance was calculated by measuring the wave velocity at room temperature at multiple wedge spacings (0.08 m to 0.14 m in 0.01 m increments) and looping through a range of plausible offset distances until the standard deviation across the range of wedge spacings was at a minimum. This ensured that the variation in measurement results was due to measurement error (e.g., small variances in setting the distance between wedges) rather than an incorrect estimation of wedge foot offset. For COMSOL, this offset value was determined by using wedge spacings of 0.075 m, 0.1 m, and 0.125 m.

### Experimental Sensitivity Analysis

The velocity calculated from time of flight measurements was affected by a number error sources. The use of 3D printed spacers ensured that the wedges were correctly aligned, and that the distance between them remained the same after removal and replacement of the wedges; however, there were still small variations in placement. The couplant used to couple the wedges to the plate allowed the wedges to slide, which increased the chance of misalignment as the couplant viscosity decreased with increasing temperature. Calculated velocity varied by ±5 m s${}^{-1}$ across multiple (30) wedge realignments. The propagation time through the wedges was subtracted from the total time of flight to leave only the time propagated through the plate. This was measured by removing the wedges from the plate and placing them together, which added another possibility of error due to misalignment. This caused a ±10 m s${}^{-1}$ variation in velocity after multiple (30) wedge realignments. The removal of the wedges from the surface of the plate to measure the wedge to wedge time allowed the wedges to cool, but care was taken to ensure that measurements were taken as quickly as possible to minimise this impact. Measuring the temperature of the plate was complicated by the temperature gradient introduced by the hot plate, which did not heat the aluminium plate evenly. Although the gradient was measured with the wedges removed, the gradient may differ slightly with the wedges present.

## 6. Results

Figure 10 shows the change in velocity with temperature for the $S_{0}$ Lamb wave mode in Aluminium 1050 H14, comparing the theoretical temperature sensitivity extracted from dispersion curves, experimental measurement data, and COMSOL simulations of the experimental system. The results from the COMSOL model were in good agreement with those taken experimentally, which also matched well the theoretical temperature sensitivity of aluminium extracted from dispersion curves. Error bars are shown for the experimental results, which show the variation across 30 calculations of velocity for each temperature. The experimental result was within 4.89 ± 2.27 m s${}^{-1}$ or 0.05% of the theoretical velocity on average. The COMSOL results were within 3.25 m s${}^{-1}$ or 0.02% of the theoretical result on average. The full dataset is provided as supplementary material, linked in the data availability statement.

As the material properties of aluminium used to generate dispersion curves were also used in the COMSOL model, an accurate model should produce similar wave velocities. The differences can therefore be attributed to the material properties of the PMMA, geometry of the wedges, and heating of the aluminium plate. The calculation of wedge foot offset accounted for variations in wedge geometry and wedge angle between the experimental system and the COMSOL model, which reduced the impact of the wedges on the measurement of plate velocity. Realistic heating of the plate and wedges in the model relied on accurate selection of heat transfer coefficients for heat flux, which were determined by experimentation, aiming to match the temperature gradients recorded during the experimental measurements. Setting these values based on material properties may yield different results.

## 7. Conclusions

This initial study shows the potential of a Lamb wave based temperature monitoring system. A COMSOL model was developed that simulated wave propagation of the $S_{0}$ Lamb wave mode in an aluminium plate using wedge transducers. The temperature of the system was raised to analyse the effect of temperature on wave propagation. The model was validated against the theoretical results extracted from dispersion curves, as well as an experimental test system upon which the model was based. Wave velocity reduced with temperature as expected, and the results were in line between the theoretical predictions and experimental results. Validating the COMSOL model now allows the model to be used to investigate, for example, the use of alternative transducer configurations, substrate materials and geometries, or the targeting of other Lamb wave modes.

In order to apply this technology to nozzle guide vanes a number of challenges need to be addressed. Curved surfaces, surface coatings, and cooling holes, will all have an effect on wave propagation, which can be investigated using the COMSOL model. The reflections produced by cooling holes may enable temperature to be monitored at a number of different locations across the surface of the vane, which is highly advantageous for identifying temperature gradients and hotspots. Different Lamb wave modes can be targeted to determine the most suitable area of the frequency-thickness spectrum for temperature monitoring applications. For permanent installation and operation at higher temperatures, an alternative transducer configuration is required. PWAS sensors, or the use of waveguides to distance the transducers from the harsh environment of a turbine, can be tested using an adapted version of the COMSOL model.

## Figures and Tables

**Figure 1 sensors-21-07390-f001:**
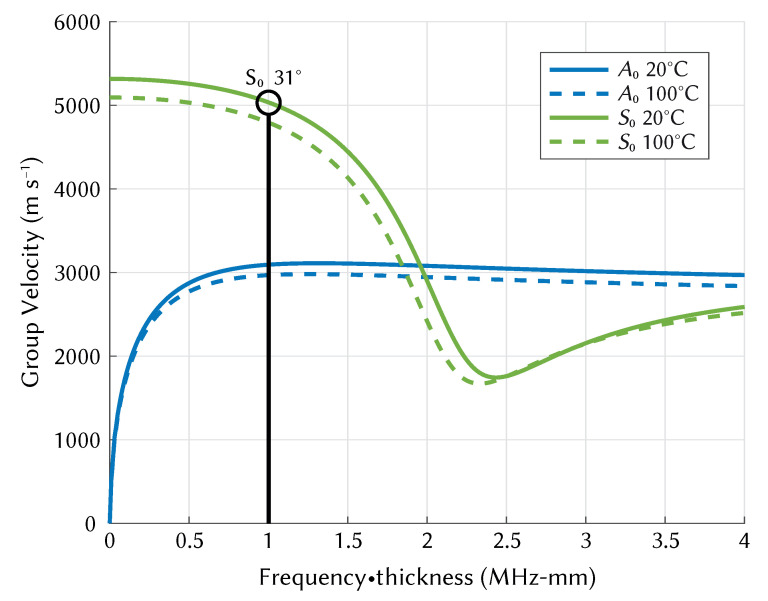
$A_{0}$ and $S_{0}$ group velocity dispersion curve shift with temperature from 20 ${}^{\circ }C$ to 100 ${}^{\circ }C$ for Aluminium 1050 H14.

**Figure 2 sensors-21-07390-f002:**
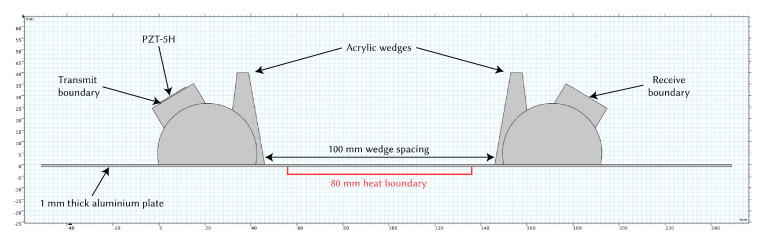
COMSOL geometry diagram.

**Figure 3 sensors-21-07390-f003:**
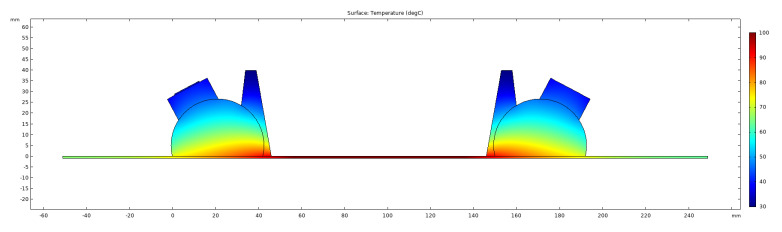
Simulated temperature gradients from stationary study at 100 ${}^{\circ }C$.

**Figure 4 sensors-21-07390-f004:**
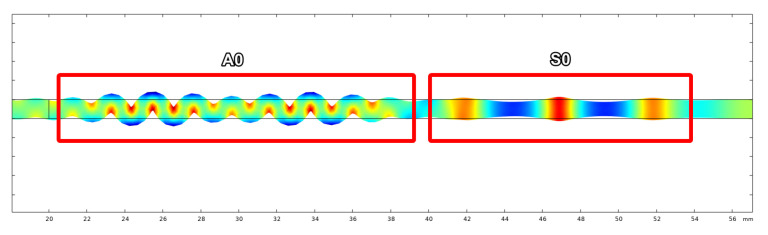
Presence of the $A_{0}$ & $S_{0}$ modes.

**Figure 5 sensors-21-07390-f005:**
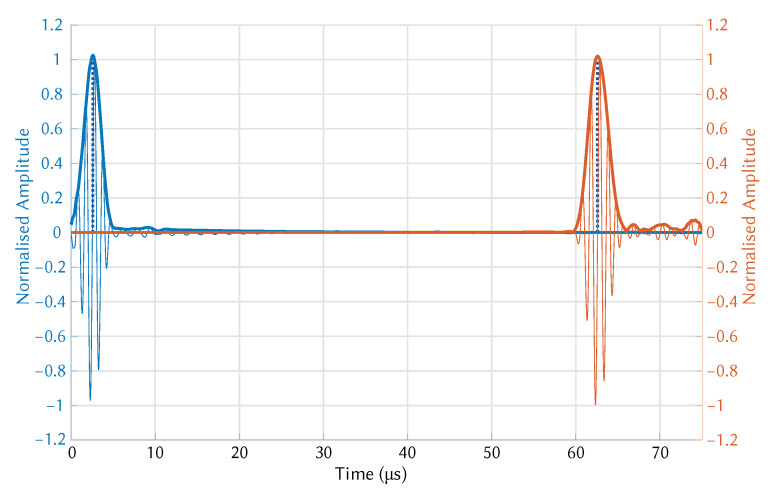
COMSOL simulation of $S_{0}$ mode propagation at room temperature.

**Figure 6 sensors-21-07390-f006:**
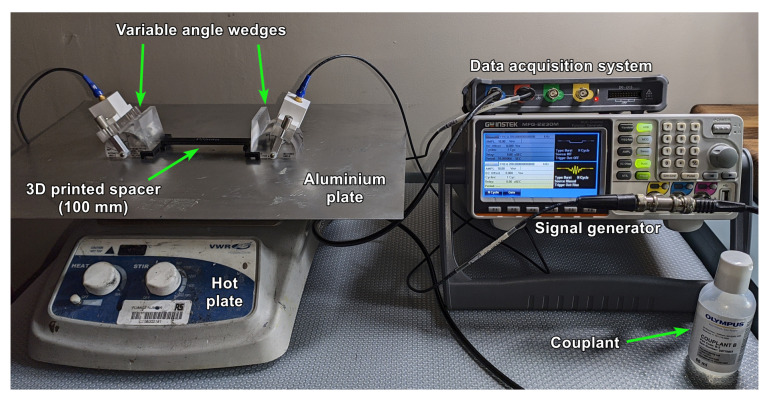
Photograph of experimental test system.

**Figure 7 sensors-21-07390-f007:**
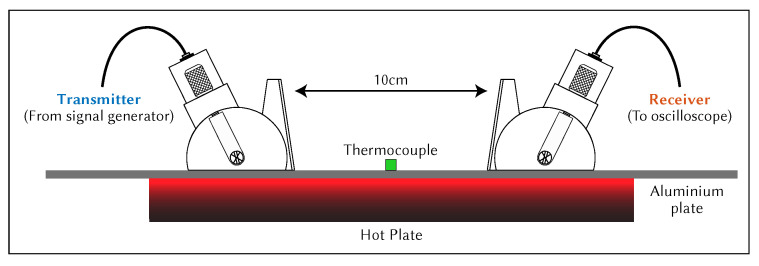
Cross-sectional diagram of total time-of-flight measurement setup.

**Figure 8 sensors-21-07390-f008:**
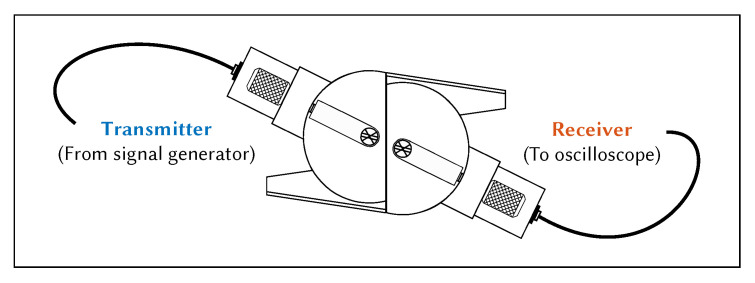
Cross-sectional diagram of wedge-to-wedge time-of-flight measurement setup.

**Figure 9 sensors-21-07390-f009:**
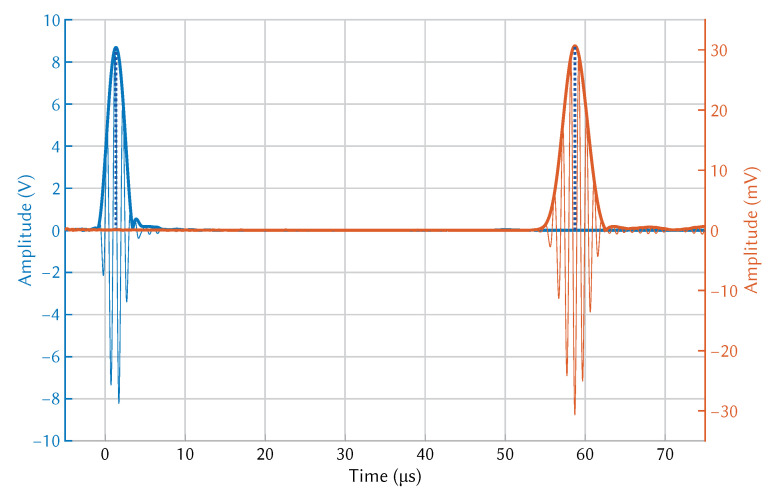
Experimentally measured wave propagation of the $S_{0}$ mode at room temperature.

**Figure 10 sensors-21-07390-f010:**
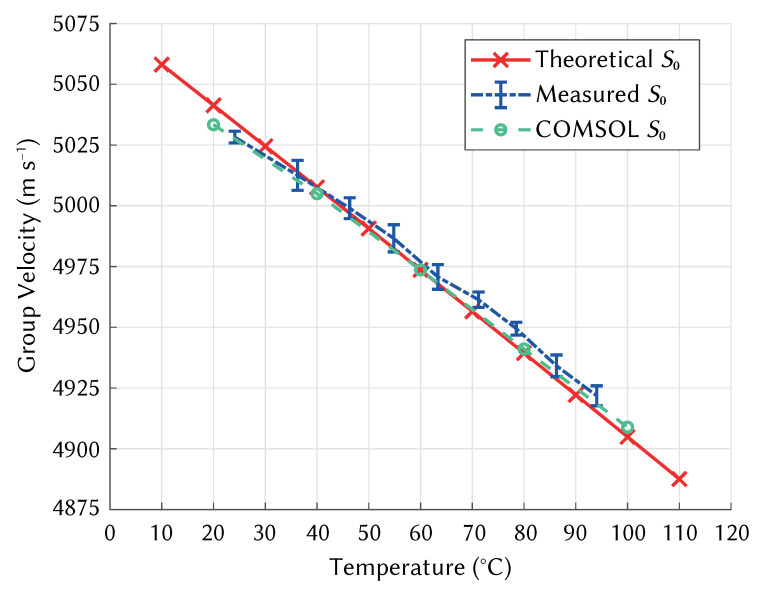
Velocity change with temperature for $S_{0}$ Lamb wave mode in Aluminium. Comparison between theoretical, experimental, and simulated results.

**Table 1 sensors-21-07390-t001:** COMSOL material properties.

Property	PMMA	Aluminium
Heat capacity at constant pressure (J/(kg·K))	1470	904
Density (kg/m${}^{3}$)	1190	2700
Thermal conductivity (W/(m·K))	0.18	237
Young’s modulus (Pa)	Equation (Equation 3)	Equation (Equation 4)
Poisson’s ratio	0.35	0.3375

**Table 2 sensors-21-07390-t002:** Experimental measurement hardware.

Measurement Hardware
2× Olympus ABWX-2001 Variable angle wedges
2× Olympus A539S-SM 1 MHz transducers
Olympus ultrasonic couplant B
GW Instek MFG-2203M Signal generator
Picoscope 3406DMSO USB Oscilloscope
Thermadata T-type temperature loggers
VWR Hot plate

## Data Availability

The data presented in this study are openly available in the University of Southampton Institutional Research Repository, ePrints Soton, at doi:10.5258/SOTON/D1963.

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
