# Peer review of "Modelling and Validation of a Guided Acoustic Wave Temperature Monitoring System"

_sensors, 2021, doi:10.3390/s21217390_

Round 1

Reviewer 1 Report

See attached file.

Author Response

Dear reviewer,

Thank you for taking the time to review our manuscript, we have carefully considered your comments and amended various parts in line with your recommendations. Point-by-point responses can be found below, and you can find the added/removed/modified parts highlighted in the attached document.

Comments:

“In the submitted manuscript the authors proposed determination of the actual temperature of nozzle guide vanes, found in the hot section of aero-engines from in situ measurements of ultrasonic velocity of the selected S0 Lamb wave mode. As a matter of fact, the idea to evaluate the temperature from of ultrasonic measurements of velocity is not new and can be traced to late 70-ties of the last century [1]. In fact, it was successfully implemented in North American nuclear power plants in on-line calibration of the thermal power of the reactor at temperatures up to 400 °C.

  • Some examples of using guided waves for temperature monitoring have been added to section 1.2 (line 90).

In general, however, the submitted manuscript is vague and contains insufficient data. As a minimum the authors should address the follow issues:

To measure the temperature the authors selected Lamb waves. Are they more sensitive to temperature changes than other types of waves, such as Rayleigh or volume wave?

  • Lamb waves are not necessarily more sensitive (their sensitivity can vary depending on material properties, mode, and frequency-thickness product, which is mentioned at the beginning of section 2) but they are less affected by surface coatings or defects than Rayleigh waves. This is mentioned on line 68. The geometry of NGVs is especially suited to Lamb wave use due to their thickness (~3 mm).

What is the expected range of in situ temperature measurements, 500, 1000, 2000 °C or higher?

  • The temperature range is limited by the ability of the transducers to operate at high temperatures. In the case of nozzle guide vanes they are operated at up to 1800C (line 119). We are considering suitable transducers for this, including distancing the transducers from the extreme temperatures using waveguides (line 167 onwards).

There is not a quantitative justification for disregarding the temperature dependencies of the Poisson's ratio, density and thermal expansion of the plate

  • We have included some quantitative justification for disregarding thermal expansion/density. Poisson ratio is considered in section 1.2 (line 110) but has now been included in the modelling section too (line 197).

The authors claim significantly high accuracy of their laboratory measurements ±0.1%. This assertion should be supported by a quantitative error (uncertainty) analysis.

  • An error analysis has been added to the end of the velocity calculation section (line 353).

The experimental part of laboratory measurements is too short. The authors have to include more
details about the experiment, such as examples of real ultrasonic signals detected with the system,
ultrasonic impulses used in TOF measurements, grade of the aluminium material used, etc.

  • Real signal plot added (fig 8). Aluminium grade added (line 304). Further detail on method included (line 315 onwards).

In real industrial conditions we always encounter the so called natural material variability, i.e., material parameters of nominally the same parts may differ by about ±1%. Consequently, employment of nominal values of the Young modulus taken from tables may lead to substantial errors in temperature evaluation from ultrasonic measurements of velocity. Can authors envisage a remedy for this issue?”

  • We agree that natural variability will play a role in determining absolute velocity. In a finalised measurement system it may be used for monitoring of changes in time-of-flight against a baseline rather than absolute temperature measurement. It may also be possible to measure the material properties directly before utilising such a monitoring system.

Reviewer 2 Report

The paper presents an interesting numerical-experimental work on the use of guided acoustics waves to monitor temperature variation of a metallic plate. The work is worthy of attention but the manuscript is not well organized and needs major revisions in order to improve its presentation and scientific impact.

  1. First of all, the order of the different sections should be changed since it forces the reader to go back and forth through the pages. The overall Section 3, concerning the Comsol simulation method, should be moved after Section 5. In my opinion, the order after paragraphs 1 and 2 should be: 3) Experimental description, 4) Velocity calculation, 5) Comsol model which includes also the paragraph 6 Results, which is too small and need to be deepened.
  2. The Introduction should be improved with a brief state of the art of structural health monitoring and its several applications. Please cite the most recent reviews on this topic.
  3. On page 2 row 61-62, the authors state that the use of guided waves for temperature monitoring has not been explored. The authors should cite the work of Salmanpour et al. in Journal of Intelligent Material Systems and Structures. 2017;28(5):604-618, doi:1177/1045389X16651155, by highlighting the differences and novelty.
  4. Pag 3, row 98: the authors should provide the names of PCBs
  5. Section 4 should be anticipated in order to understand more easily the description of Comsol model
  6. Experimental pag 7-8: provide more info on the model and characteristics of acrylic wedges (max Temperature), transducers (nominal frequency, diameter, maximum operating temperature, etc.) on the couplant (maximum operating temperature). These details can be added in Table 2 or in the text.
  7. Page7 , row 265:Provide the measure unit to sampling rate
  8. Page 8, row 270: Since Ref 29 is not freely accessible, please provide more details on the test method.
  9. Which angles have been used to angulated the transmitting and receiving transducers? How did the authors choose this angle?
  10. How the temperature is raised in the experimental set-up?
  11. Section 6 Results is very small. Figure 9 is poorly discussed. First of all, the way used to predict the theoretical results is not clear. The authors refer to The Dispersion Calculator [15] without any further detail. It is very difficult to verify their data.
  12. The experimental measurement of group velocity should be better presented. The authors should add the plot with the echoes at each temperature.
  13. Figure 5: the authors should better explain the meaning of the blu and cred peak.

Author Response

Dear reviewer,

Thank you for taking the time to review our manuscript, we have carefully considered your comments and amended various parts in line with your recommendations. Point-by-point responses can be found below, and you can find the added/removed/modified parts highlighted in the attached document.

Comments:

The paper presents an interesting numerical-experimental work on the use of guided acoustics waves to monitor temperature variation of a metallic plate. The work is worthy of attention but the manuscript is not well organized and needs major revisions in order to improve its presentation and scientific impact.

First of all, the order of the different sections should be changed since it forces the reader to go back and forth through the pages. The overall Section 3, concerning the Comsol simulation method, should be moved after Section 5. In my opinion, the order after paragraphs 1 and 2 should be: 3) Experimental description, 4) Velocity calculation, 5) Comsol model which includes also the paragraph 6 Results, which is too small and need to be deepened.

  • The thinking behind the current structure is that the focus is on the COMSOL model, so the experimental section is only included later on for verification purposes. The velocity calculation section follows these sections because it is only after the data is collected that the velocity can be calculated (and the same methods apply to both experimental and simulated results). I have included more links between sections to allow for faster navigation. We would prefer not to change this.

The Introduction should be improved with a brief state of the art of structural health monitoring and its several applications. Please cite the most recent reviews on this topic.

  • We agree, this has now been included in section 1.

On page 2 row 61-62, the authors state that the use of guided waves for temperature monitoring has not been explored. The authors should cite the work of Salmanpour et al. in Journal of Intelligent Material Systems and Structures. 2017;28(5):604-618, doi:1177/1045389X16651155, by highlighting the differences and novelty.

  • There are many studies that consider temperature compensation techniques (such as that by Salmanpour et al.) but a very limited number that consider utilising Lamb wave propagation through a structure for temperature measurement/monitoring. A number of studies have now been included however (line 90).

Pag 3, row 98: the authors should provide the names of PCBs

  • This has been changed to “printed circuit boards (PCBs)”.

Section 4 should be anticipated in order to understand more easily the description of Comsol model

  • We prefer to cover the COMSOL model first as this is the focus of the study, the experimental system is included only for validation purposes.

Experimental pag 7-8: provide more info on the model and characteristics of acrylic wedges (max Temperature), transducers (nominal frequency, diameter, maximum operating temperature, etc.) on the couplant (maximum operating temperature). These details can be added in Table 2 or in the text.

  • This section has been updated to provide more information.

Page7 , row 265:Provide the measure unit to sampling rate

  • “Hz” added.

Page 8, row 270: Since Ref 29 is not freely accessible, please provide more details on the test method.

  • Additional details added.

Which angles have been used to angulated the transmitting and receiving transducers? How did the authors choose this angle?

  • The angle is given in equation 1 & 2.

How the temperature is raised in the experimental set-up?

  • A hot plate (line 312).

Section 6 Results is very small. Figure 9 is poorly discussed. First of all, the way used to predict the theoretical results is not clear. The authors refer to The Dispersion Calculator [15] without any further detail. It is very difficult to verify their data.

  • A link to Table 1 has now been provided which gives the material properties required to reproduce the dispersion curves using the dispersion calculator. All data can be verified with the datasets and code provided in the repository linked in the data availability statement (this will go live upon publication).

The experimental measurement of group velocity should be better presented. The authors should add the plot with the echoes at each temperature.

  • An example of wave propagation at room temperature has been added (fig 8).

Figure 5: the authors should better explain the meaning of the blue and red peak.

  • Additional information has been added to the text for both the simulated and experimental plots. The thick blue and orange lines indicate the envelope of the signals, while the blue dotted lines indicate the peak of the envelopes. It is these points that are used to calculate the time-of-flight.

Round 2

Reviewer 2 Report

Even if my suggestion to change the paragraph order has not been addressed, the authors have considerably improved the paper. It can be published now